# Electrochemical synthesis of biaryls by reductive extrusion from *N,N'*-diarylureas

Ellie Stammers[1], Chris D. Parsons[2], Jonathan Clayden [1] ✉ & Alastair J. J. Lennox [1] ✉

The synthesis of biaryl compounds by the transition-metal free coupling of arenes is an important contemporary challenge, aiming to avoid the toxicity and cost profiles associated with the metal catalysts commonly used in the synthesis of these pharmaceutically relevant motifs. In this paper, we describe an electrochemical approach to the synthesis of biaryls in which aniline derivatives are coupled through the formation and reduction of a temporary urea linkage. The conformational alignment of the arenes in the *N,N'*-diaryl urea intermediates promotes C-C bond formation following single-electron reduction. Our optimized conditions are suitable for the synthesis of a variety of biaryls, including sterically hindered examples carrying ortho-substituents, representing complementary reactivity to most metal catalysed methods.

The biaryl motif is a feature found in many compound classes, including natural products, pharmaceuticals, agrochemicals, organic electronic devices, and ligands for metals[1–3]. The limited conformational flexibility provided by the biaryl linkage makes it a 'privileged' moiety in bioactive molecules, prevalent in drugs across a range of therapeutic areas[4].

Transition metal-catalysed cross-coupling methods provide the most common synthetic approaches to biaryls[5–7], especially within medicinal chemistry[8]. These methods have the advantage of broad scope and versatility but set against this, the continuity of supply, the cost, and the toxicity of the transition metal catalysts have prompted significant exploration of alternative, metal-free, cross-coupling strategies[9]. Challenges associated with this class of reaction include regioselectivity[10], functional group tolerance[11], very specific coupling partners[12–18], or a requirement for a high excess of coupling partner, often up to solvent level[19–25]. Many of these properties may be alleviated by an alternative intramolecular strategy in which both arenes are contained within the same species. Prominent examples of such precursors include phosphonium salts that allow the coupling of two heteroaromatic rings[26–30], as well as sulfoxides[31,32], sulfuranes[33–37], sulphonamides[38–41] and boron[42–46] complexes (Fig. 1A).

Intramolecular reactivity is enhanced by structural features that favour reactive conformations, classically formulated as the Thorpe-Ingold effect. Thus, intramolecular nucleophilic aromatic substitution reactions of *N*-aryl ureas carrying a nucleophilic *N'*-substituent are accelerated by their conformational preference for the *N*-aryl ring to lie trans to the urea carbonyl group, and hence cis to the nucleophile (Fig. 1B). In this way, even electron-rich aniline derivatives exhibit electrophilic reactivity towards anionic nucleophiles, providing versatile methods for C(sp³)-arylation[47].

Stevenson and co-workers reported in 2003 that reduction of *N,N*-dimethyl-*N,N*-diphenylurea using sodium in HMPA generates a biaryl radical anion, observable by EPR[48,49]. This spectroscopic observation suggested that reductive formation of a radical anion from an *N*-aryl urea could be used as a general trigger for Ar−Ar bond formation, with the return of aromaticity leading to C−N bond cleavage and extruding biaryl as an isolable product (Fig. 1C).

In this work, we show the development of a preparative method for the synthesis of biaryls by reductive cross-coupling based on this hypothesis. We show that the well-established conformational preference of *N,N*-dialkyl-*N,N*-diarylureas[50–54] can be used to promote a transition-metal-free aryl-aryl cross-coupling reaction, which is complementary to existing biaryl forming strategies.

## Results and discussion

Lithium metal, in combination with di-*tert*-butylbiphenyl (DBB), has proved successful for the reduction of arenes under ammonia-free Birch conditions[55] and is more practical than sodium in HMPA. Using

[1]School of Chemistry, University of Bristol, Cantock's Close, Bristol BS8 1TS, UK. [2]Early Chemical Development, Pharmaceutical Sciences, R&D, AstraZeneca, Macclesfield SK10 2NA, UK. ✉e-mail: j.clayden@bristol.ac.uk; a.lennox@bristol.ac.uk

**A** Metal-free biaryl formation by intramolecular coupling

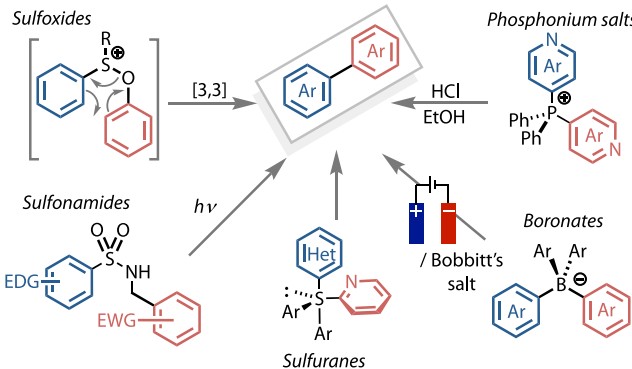

**B** Urea promotes intramolecular C-C coupling

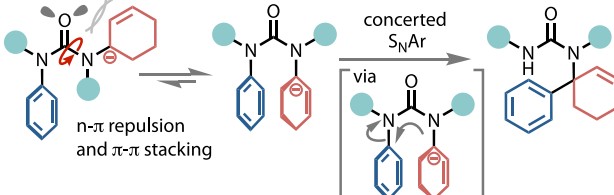

**C** *Hypothesis*: Reduction of *N,N′*-diaryl ureas allows synthesis of biaryls

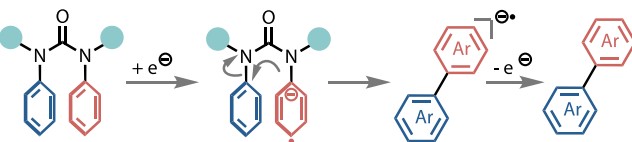

**Fig. 1 | Previous intramolecular coupling strategies and project hypothesis.**
**A** Overview of transition metal-free biaryl coupling involving an intramolecular C–C bond-forming step. **B** Urea facilitates unconventional intramolecular C–C coupling from cis conformation. **C** Reductive extrusion of biaryls from reduced diaryl ureas.

## Table 1 | Reaction optimisation

| | Reduction conditions[a] {−/+} | LiCl eq. | 1a/%[b] | 2a/%[a] |
|---|---|---|---|---|
| 1 | Li, DBB | 0 | 0 | 10 |
| 2 | −6 mA {Gr/Gr} | 0 | 27 | 51 |
| 3 | −6 mA {Gr/Gr} | 5 | 0 | 86 |
| 4 | −6 mA {Gr/Gr} | 10 | 0 | 77 |
| 5 | −6 mA {Pt/Gr} | 5 | 0 | 54 |
| 6 | −6 mA {GC/Gr} | 5 | 10 | 54 |
| 7 | −6 mA {Gr/Pt} | 5 | 0 | 79 |
| 8 | −6 mA {Gr/CC} | 5 | 13 | 68 |
| 9 | −6 mA {Gr/Gr} in DMSO | 5 | 0 | 67 |
| 10 | −6 mA {Gr/Gr} in MeCN | 5 | 18 | 44 |

[a]Electrochemical reactions conducted at constant current with {cathode/anode}. *Gr* graphite, *CC* carbon cloth.
[b]¹H NMR yields.

this method for reduction of a model unsymmetrical urea substrate **1a**, variation of solvents, concentration, and reagent quantity ratios gave **2a** (entry 1), but in a yield of only 10%. Other substrates were surveyed, most of which gave similarly low yields, except for the electron-deficient *N,N′*-bis(4-cyanophenyl)urea **1b** (see SI).

Despite promising indications that reductive coupling was possible, the poor yield with LiDBB led us to explore alternatives. Among these, electrochemical reduction offers the advantage of tuneable redox potential and is inherently sustainable[56–59], avoiding the use of stoichiometric metal. We thus turned to the electrochemical reduction of **2a**, optimising applied current, electrode material, solvent, and concentration. We noted an instant improvement with the cathodic reduction of the urea, giving a 51% yield of **2a**. After optimisation of other variables, including the addition of LiCl as an additive, an improved yield of 86% was achieved. The reaction outcome was particularly sensitive to the solvent, the amount of LiCl, and the electrode materials (Table 1)[60]. Sacrificial anodes such as aluminium worked well, but we chose to avoid the production of stoichiometric metal waste through the oxidation of bromide (present in the electrolyte as Bu₄NBr) to tribromide. We propose that this method works particularly well because of the very slow migration of anionic tribromide product to the cathodic chamber and avoiding competitive reduction[61].

The conformational preference imposed by the dimethylurea linker, which we propose brings the coupling partners into proximity, is evident in the ¹H NMR spectra of **3b** and **1b**. An upfield shift of the aromatic ¹H NMR signals on moving from monoaryl urea **3b** to diarylurea **1b** (Fig. 2A) suggests shielding of the proximal p-system. The role of this conformational preference on reactivity was tested by

comparing the reduction of the dimethyl urea **1b** with the corresponding cyclic analogue **4b**, in which the aryl substituents necessarily point away from each other (Fig. 2B). None of the coupled **2b** was observed, consistent with our proposal. Additional evidence for an intramolecular, as opposed to intermolecular, coupling was provided by the lack of homo-coupled products from unsymmetrical ureas (see below) and by the absence of product when monoaryl urea **3b** was subjected to the reaction conditions. Molecules containing other linker units were also prepared and tested in order to gauge their reactivity relative to dimethylurea. Ethyl substitution (**5b**) was tolerated, but any further changes were detrimental to the reaction outcome. The carbamate (**6b**) and thiourea (**7b**) gave low yields, and the unmethylated urea (**8b**) and sulphonamide (**9b**) afforded no product. With substrates other than ureas, significant cleavage of the linker was observed under the reaction conditions (see SI for more details).

Rapid biaryl formation on the cyclic voltammetry (CV) timescale was evident from CV analysis of **1a**, Fig. 3A. The CV trace of **1a** was irreversible, indicating that the radical anion formed on the electrode surface must react to give further products. Indeed, the reverse sweep shows an oxidation peak for product **2a**, and the second scan shows a new reduction peak for product **2a** formed in the first scan. In contrast, the monoaryl urea **3b** gave a reversible CV trace, indicating that, without a second arene coupling partner, no onward reaction takes place from the reduced species. Product **2a** was formed on the CV timescale at scan rates up to 1 V/s, the fastest tested (Fig. 1B). The observed linear relationship between the $I_{pc}$ and the square root of the scan rate reveals, according to the Randles-Sevcik equation, freely diffusing redox species. However, the reduction peak shape changed when LiCl was added. When LiCl was titrated, the peak adopted a progressively more symmetrical shape, with no evidence for the formation of **2a**. A loss of current was observed in subsequent scans when the electrode was not polished. Although this CV analysis could not fully reveal the role of LiCl, an electron-transfer process of a surface-bound LiCl–urea adduct could account for this evidence (see SI for details)[62,63].

In light of these results, we propose the reaction mechanism illustrated in Fig. 3C, in which the key step is the intramolecular C–C

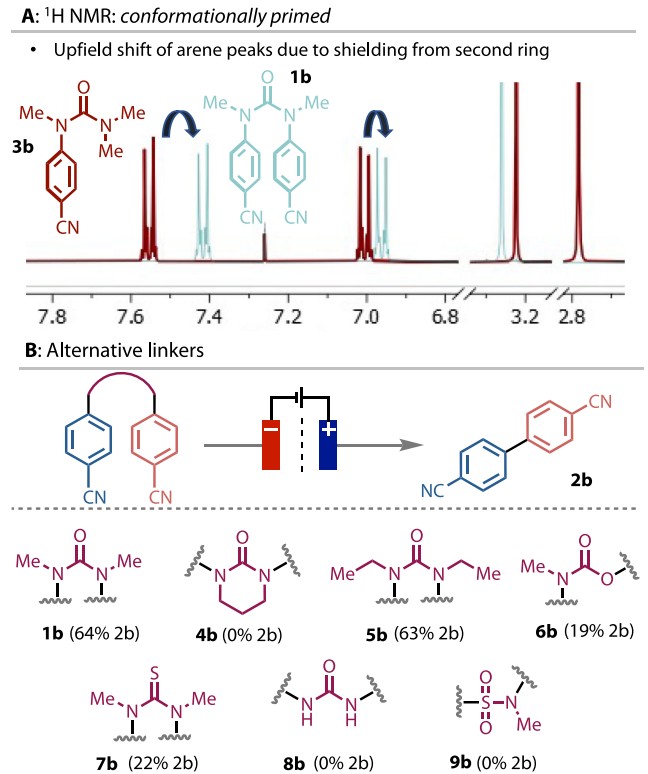

**Fig. 2 | Conformation of diarylureas and variation of alternative linkers. A** ¹H NMR of **1b** and **3b**, which demonstrates the shielding effect of the second ring in **1b** due to the cis conformation. **B** A selection of alternative linkers that were tested in the biaryl coupling reaction.

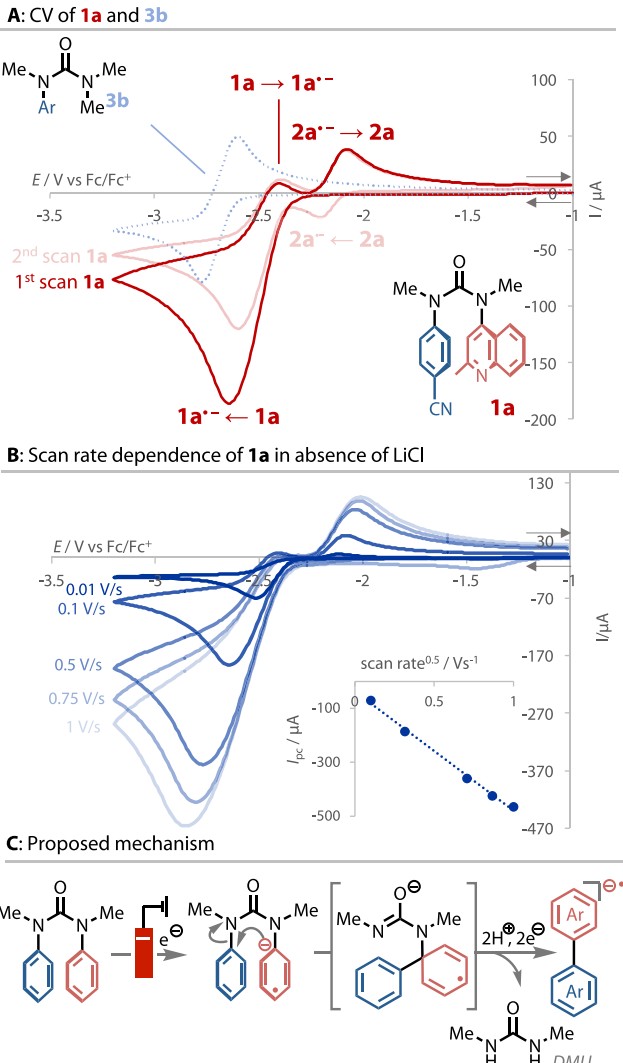

**Fig. 3 | Mechanistic studies. A** CV of **1a** with 1 repeat scan, co-plotted with a CV of **3b**. Glassy carbon (−)|Pt wire (+), Ag/AgNO₃ (ref), 0.1 V s⁻¹, 0.025 mmol substrate, 0.1 M TBAPF₆, 0.05 M DMF; **B** scan rate dependence of **1a**, inset: cathodic peak current vs scan rate⁰·⁵; **C** proposed overall mechanism, DMU dimethylurea.

coupling of the conformationally preorganised reduced urea. Considering the electronic promiscuity of the reaction (see below), it is likely that a concerted $S_NAr$ reaction occurs, as proposed in related reactions of *N*-aryl ureas[64–66]. A time-course experiment showed that dimethyl urea (DMU) is the sole by-product of the reaction and is formed in direct parity with product **2b** (see SI). Several possibilities exist for the liberation of the biaryl from the rearranged intermediate, including C-N bond homolysis, a second reduction, or the intermediate formation of a diaziridinone[48,67]. The biaryl appears to persist in a reduced state (a radical anion or dianion) until work-up, at which point it undergoes aerobic oxidation. Indeed, we noted the decomposition of these reduced species if the reaction mixture was left for prolonged periods without workup.

Figure 4 details the range of biaryl products that have been formed by electrochemical reduction of *N,N'*-diarylureas. Using the same conditions (conditions a) as model substrates **2a** and **2b**, *ortho* alkyl substitution was well tolerated in reactions giving biaryls **2c–e**, suggesting that steric hindrance was not detrimental to the coupling reaction.

Trifluoromethyl and fluoro substituents were also tolerated (**2f, 2g**), though these rings coupled in lower yield. Unsymmetrical biaryls were similarly formed when a *p*-cyanophenyl ring was paired with other electron-deficient rings, yielding products **2h–2k**, and the coupling of nitrile or ester-substituted rings (**2h, 2j**) were the highest yielding, with nitro substituted **2i** the lowest yielding presumably due to enhanced stability of the reduced form. Coupling *p*-cyanophenyl with biphenyl (**2l**) and naphthalene (**2m, 2n**) partners having more extended, and therefore reducible and electrophilic, π-systems was also especially effective.

Coupling with more electron-rich partners required a change of reaction conditions (conditions b). Platinum electrodes were used, as the graphite electrodes were unstable at very deep electrode

potentials, and LiCl was removed to avoid competitive reduction of lithium cations. These modifications enabled coupling of the *p*-cyanophenyl ring to phenyl (**2o**) and ortho-substituted phenyl (**2p, 2q**) and fluorophenyl (**2r**) rings, as well as with methoxy and amino-substituted partners (**2t, 2v** and **2w**). Limitations to the tolerance of less electron-withdrawing and halogen substituents were indicated by the failure of the 4-methoxyphenyl coupling (**2u, 2s**), in contrast to the successful 3-methoxyphenyl coupling **2t**. Such substituents are nonetheless tolerated in the 4-position (**2r, 2w**) provided they are accompanied by a 2-alkyl substituent, an observation that highlights a beneficial effect from 2-alkyl substituents that is also apparent when comparing the yields of **2o** with **2p** and **2q**, and in the formation of **2af**. *Ortho* substituents may help to increase the population of the reactive conformer by favouring face-to-face rather than edge-to-edge p-p interactions. This effect contrasts with established cross-coupling methods, in which more hindered partners require more reactive and specialised catalyst systems[68–71].

Successful coupling with more electron-deficient and conjugated p-systems was extended further to ureas in which neither ring carried a *p*-cyano group. The biphenyl-4,4'-diester **2x**, biquinolyl **2y**, binaphthyl **2z** and even the hindered atropisomeric

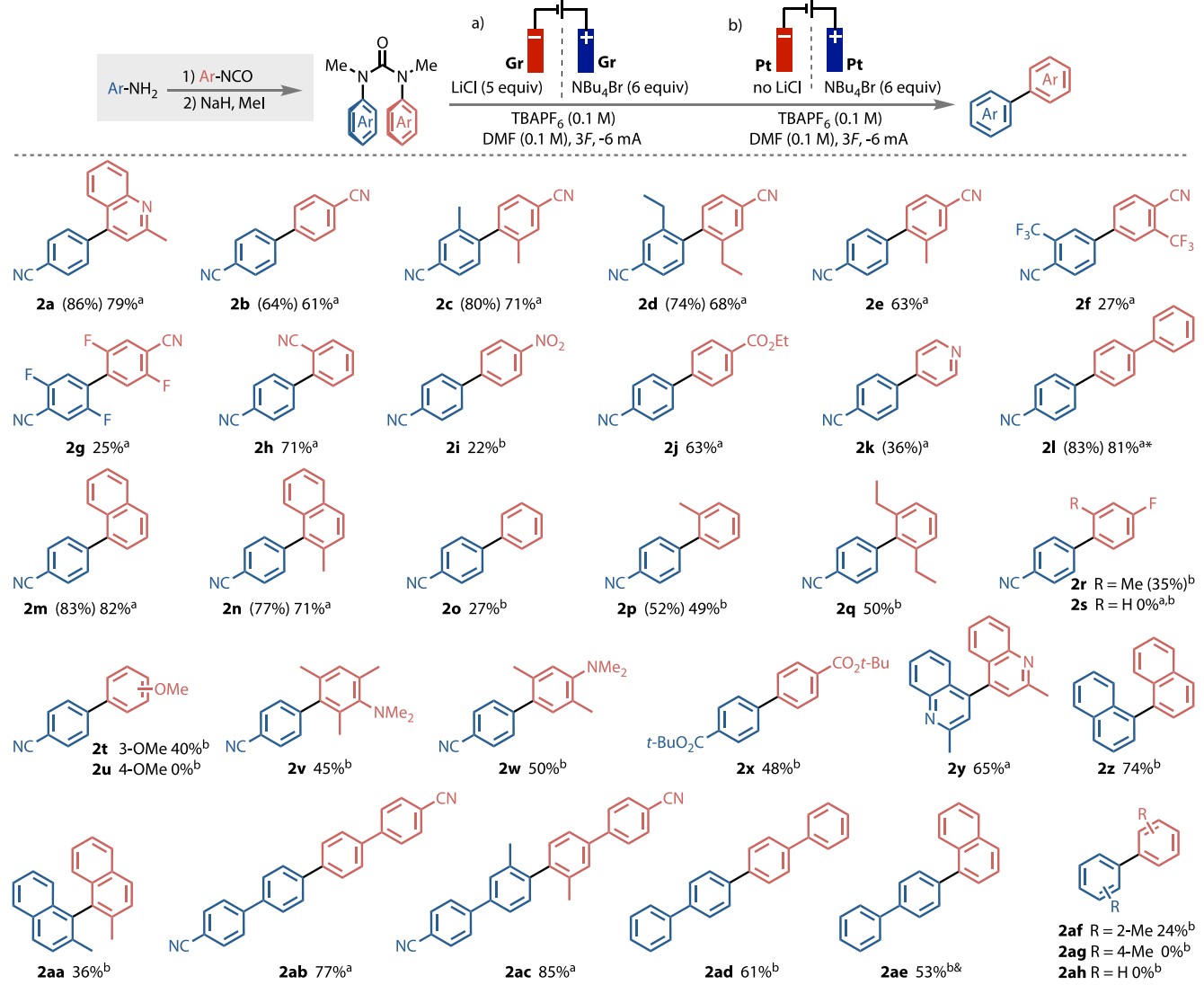

**Fig. 4 | Substrate scope.** Substrate scope, Yields = isolated yields (NMR yields); Gr = graphite rod. *Gr (−) flaked during reaction, & 2F applied.

tetra-ortho-substituted 2,2'binaphthyl **2aa** all formed success-fully. Biphenyl substrates coupled exceptionally well, giving tet-raphenyls **2ab**, **2ac** and **2ad**, and naphthylbiphenyl **2ae**. Biphenyl itself could not be formed from *N,N'*-diphenylurea, but 2,2'-dimethylbiphenyl **2af** was formed in low yield.

In summary, we report a new electrochemical reductive method for the formation of biaryls. This metal-free approach uses readily available anilines as coupling partners, tethering them through a urea linkage whose conformational preference enforces proximity between the arene rings. Electrochemical reduction forms the biaryl product, extruding dimethylurea as a by-product. The reaction scope is com-plementary to more established transition-metal catalysed coupling methods, being especially amenable to electron-deficient and steri-cally hindered ortho-substituted biaryl products.

## Methods

The Supplementary Information provides full details of methods for the synthesis of all urea starting materials, their spectroscopic char-acterisation and their reduction to biaryls. For the electrochemical reduction of urea to a biaryl: a solution of the urea (1 eq), LiCl (5 eq), TBAPF$_6$ (0.1 M) in DMF on the cathodic side of a divided cell and TBAPF$_6$ (0.1 M) and TBAB (6 eq) in DMF on the anodic side were

electrolysed (3*F*, −6 mA) with graphite electrodes under an atmo-sphere of N$_2$.

## Data availability

The data that support the findings of this study are available in the supplementary material of this article.

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

## Acknowledgements
The authors acknowledge funding from the Royal Society (University Research Fellowship and Enhancement Awards to A.J.J.L.), the ERC (Advanced Grant 883789 to J.C.), EPSRC (EP/S018050/1 and EP/L015366/1) and AstraZeneca for a studentship (to E.S.) through the Bristol Centre for Doctoral Training in Chemical Synthesis.

## Author contributions
E.S. conducted all experimental work, J.C. and A.J.J.L. conceived and directed the research project, C.D.P., J.C., and A.J.J.L. supervised the project and sourced funding. All authors contributed to writing the paper.

## Competing interests
The authors declare no competing interests.
