## [Peer Review File · Nature Communications]

REVIEWER COMMENTS

Reviewer #1 (Remarks to the Author):

In this work, Lennox and Clayden described an electrochemical approach to the synthesis of biaryls by the reduction of N,N'-diaryl urea intermediates. The work presents a new synthetic method with well-presented mechanistic insight and a reasonably wide substrate scope. I recommend its publication as it is.

Reviewer #2 (Remarks to the Author):

The authors describe an electrochemical approach to the synthesis of biaryls in which aniline derivatives are coupled through the formation and reduction of a temporary urea linkage. This work provides a novel method for the synthesis of biaryls by electroreduction of the intramolecular reaction process. However, there are the following problems. I think this manuscript is not suitable for its publication in Nature Commun.

(1) This work did not demonstrate good functional group tolerance and reasonable generality in substrate scope. The authors list more than 30 substrates (2a-2ah), but most of them contain cyanogen groups and the synthesis of some symmetrical diaryl compound (2x-2ad).

(2) A few mechanistic experiments cannot adequately support the proposed mechanism. Why is it that instead of continuing to have one electron to form an anion to break the C-N bond, the transition state takes two electrons to form a radical anion? Can you prove the radical anion?

(3) Some text and formatting corrections. For example, 9b should be changed to 3b (line 30); 6 mA formatting error (table 2).

Reviewer #3 (Remarks to the Author):

The manuscript proposed by Clayden and Lennox presents a catalyst-free coupling of biaryl derivatives under electrochemical reductive conditions. It provides an elegant conceptual transition

metal free alternative to classical cross-coupling reaction, based on the reduction of diarylureas. This study is very well conducted, from the optimization of reaction conditions to the evaluation of the substrate scope. It also provides plenty of useful experimental data to support the proposed mechanism.

The authors finally demonstrated the possibility of coupling aryl partners of similar electronic properties, that can be quite challenging for tetraarylborates under oxidative conditions.

It seems that proximity between the two aryl substituents is essential to achieve the coupling reaction, and that therefore an intramolecular process should be largely favoured. Nevertheless, did the authors try crossover reactions by mixing two different starting ureas? This would be important to definitely rule out the possibility of intermolecular couplings.

The authors did not disclose the case of nitrated product 2i (lowest yield of 22%). Is the nitro group problematic in this transformation, does it undergo reduction under the given conditions?

The concept presented here definitely deserves to be published in a high ranking journal such as Nat. Commun., and I suggest its acceptance after the authors answered above comments and suggestions.

Quotations from the reviewers are copied in their entirety and provided in *blue italicized font*.

Reviewer 1:

In this work, Lennox and Clayden described an electrochemical approach to the synthesis of biaryls by the reduction of N,N'-diaryl urea intermediates. The work presents a new synthetic method with well-presented mechanistic insight and a reasonably wide substrate scope. I recommend its publication as it is..

We'd like to thank this reviewer for their careful reading of the manuscript and for their positive comments.

Reviewer 2:

The authors describe an electrochemical approach to the synthesis of biaryls in which aniline derivatives are coupled through the formation and reduction of a temporary urea linkage. This work provides a novel method for the synthesis of biaryls by electroreduction of the intramolecular reaction process. However, there are the following problems. I think this manuscript is not suitable for its publication in Nature Commun.

We are happy this reviewer acknowledges the novelty of the method, as this is the most important point – it is a completely new way to make biaryls that has different characteristics and features compared to other methods.

(1) This work did not demonstrate good functional group tolerance and reasonable generality in substrate scope. The authors list more than 30 substrates (2a-2ah), but most of them contain cyanogen groups and the synthesis of some symmetrical diaryl compound (2x-2ad).

While many of the biaryls do indeed contain nitrile groups, many of them do not. What is especially interesting about the scope is that it is particularly tolerant of ortho-substitution. This is a feature that other coupling methods, such as metal-catalysed processes, are very poor at tolerating. So, although our method is less tolerant of electron donating groups (although we do have some!), it is better with other features that other methods are less good at. Hence, we think it is important to emphasise the complementarity of the method to other methods.

(2) A few mechanistic experiments cannot adequately support the proposed mechanism. Why is it that instead of continuing to have one electron to form an anion to break the C-N bond, the transition state takes two electrons to form a radical anion? Can you prove the radical anion?

Thank you for posing this question. We do believe we are correctly interpreting the data to inform our mechanistic proposal. The cathode is injecting 1 electron into the neutral starting material, which will generate a radical anion. This radical anion then does the rearrangement. This process can be observed in the CV. The further reduction to the dianion occurs at much more negative potentials. The radical anion of the product gets oxidised on work up to give the neutral product.

If the reviewer is referring to the next step, which is about how the rearranged radical anion then decomposes to form the radical anion of the product and DMU, then we agree that it is not clear what the mechanism of this is, and this is exactly why we have not inferred a particular process for that step. We have made the following proposals for this step in the text but have been deliberately careful:

“Several possibilities exist for the liberation of the biaryl from the rearranged intermediate, including C-N bond homolysis, a second reduction, or the intermediate formation of a diaziridinone.^{[17,24]”}

The CV tells us that the product is more easily reduced than the starting material. With this information and the fact that we have coloured solutions, we know the radical anion of the product

is forming. Then, as we also know we are forming DMU from proton NMR analysis of reaction mixtures (See SI for details), it takes another 2 electrons and 2 protons to make that product. Therefore, we are just giving the balanced equation for the process.

(3) Some text and formatting corrections. For example, 9b should be changed to 3b (line 30); 6 mA formatting error (table 2).

Thank you for spotting these errors. They have now been fixed.

Reviewer 3:

The manuscript proposed by Clayden and Lennox presents a catalyst-free coupling of biaryl derivatives under electrochemical reductive conditions. It provides an elegant conceptual transition metal free alternative to classical cross-coupling reaction, based on the reduction of diarylureas. This study is very well conducted, from the optimization of reaction conditions to the evaluation of the substrate scope. It also provides plenty of useful experimental data to support the proposed mechanism.

The authors finally demonstrated the possibility of coupling aryl partners of similar electronic properties, that can be quite challenging for tetraarylborates under oxidative conditions.

We'd like to thank the reviewer for these positive comments and their careful reading of the manuscript.

It seems that proximity between the two aryl substituents is essential to achieve the coupling reaction, and that therefore an intramolecular process should be largely favoured. Nevertheless, did the authors try crossover reactions by mixing two different starting ureas? This would be important to definitely rule out the possibility of intermolecular couplings.

Thank you for this insightful comment. Indeed, this experiment has been conducted by monitoring the product distribution when using an unsymmetrical urea. The hypothesis was that if homocoupled products were observed then intermolecular coupling had to be occurring. However, we never observed any homo-coupling. These experiments are detailed in the SI and also in the manuscript (line 18, page 3).

The authors did not disclose the case of nitrated product 2i (lowest yield of 22%). Is the nitro group problematic in this transformation, does it undergo reduction under the given conditions?

Thanks for this interesting question. Indeed, highly electron withdrawing functional groups such as nitro groups are readily reduced and the anionic products formed are more stable, and less inclined to do the coupling. We have added in the following sentence to the manuscript to reflect this good point:

“with nitro substituted 2i the lowest yielding presumably due to enhanced stability of the reduced form”

The concept presented here definitely deserves to be published in a high ranking journal such as Nat. Commun., and I suggest its acceptance after the authors answered above comments and suggestions.

Thank you for this positive evaluation.